# An exact mapping between loop-erased random walks and an interacting field theory with two fermions and one boson

Assaf Shapira[1] and Kay Jörg Wiese[2]

**1** Dipartimento di Matematica e Fisica, Università Roma Tre,
Largo S.L. Murialdo, 00146, Roma, Italy.
**2** Laboratoire de Physique de l'École Normale Supérieure, ENS,
Université PSL, CNRS, Sorbonne Université, Université Paris-Diderot,
Sorbonne Paris Cité, 24 rue Lhomond, 75005 Paris, France.

## Abstract

We give a simplified proof for the equivalence of loop-erased random walks to a lattice model containing two complex fermions, and one complex boson. This equivalence works on an arbitrary directed graph. Specifying to the $d$-dimensional hypercubic lattice, at large scales this theory reduces to a scalar $\phi^4$-type theory with two complex fermions, and one complex boson. While the path integral for the fermions is the Berezin integral, for the bosonic field we can either use a complex field $\phi(x) \in \mathbb{C}$ (standard formulation) or a nilpotent one satisfying $\phi(x)^2 = 0$. We discuss basic properties of the latter formulation, which has distinct advantages in the lattice model.



# 1 Introduction

The loop-erased random walk (LERW) introduced by Lawler in [1] and further developed in [2–4] can in two dimensions be described by Schramm-Löwner evolution (SLE) at $\kappa = 2$ [5]. It corresponds to a conformal field theory with central charge $c = -2$. When reformulated in terms of loops, the lattice $\mathcal{O}(n)$-model can be defined for $-2 \leq n \leq 2$ [6] (see also [7], page 187). In the limit of $n \to -2$ it has the same conformal field theory (CFT), a relation which also holds off criticality [8], suggesting the description by one complex fermion. The latter, however, does not allow one to assess properties of loop-erased random walks after erasure. Recently, it was established [9, 10] that the equivalence to the $\mathcal{O}(n)$ model holds in any dimension $d \geq 2$, by mapping loop-erased random walks onto the $n$-component $\phi^4$-theory for $n \to -2$, or equivalently a theory with two complex fermions, and one complex boson. This formulation contains information about the traces of LERWs after erasure, and in particular its Hausdorff dimension.

A rigorous proof of the equivalence between LERWs and the $\mathcal{O}(-2)$ model was given in Ref. [11]. The ensuing field theory uses *nilpotent* bosonic fields, i.e. fields which square to zero. Here we give a simplified proof of the equivalence. We also show that the nilpotent bosonic fields can be replaced by standard bosonic ones. The ensuing field theory has two complex fermions, and one (standard) complex boson, and allows to evaluate, without approximations, observables defined on an arbitrary graph. Keeping only the most relevant terms, it reduces to the $\phi^4$-type theory with two fermions and one boson given in Refs. [9,10]. Interestingly, there this formulation arose naturally in the analysis of charge-density waves subject to quenched disorder. As a result, the theories discussed here are linked to such diverse systems as charge-density waves, Abelian sandpiles [12, 13], uniform spanning trees [14, 15], the Potts model [14], Laplacian walk [2, 16], and dielectric breakdown [17].

# 2 The loop-erased random walk

Consider a random walk on a directed graph $\mathcal{G}$. The walk jumps from vertex $x$ to $y$ with rate $\beta_{xy}$, and dies out with rate $\lambda_x = m_x^2$. The coefficients $\{\beta_{xy}\}_{x,y \in \mathcal{G}}$ are weights on the graph. In particular, when $\beta_{xy}$ is positive, $\mathcal{G}$ contains an edge from $x$ to $y$. Denote by $r_x = \lambda_x + \sum_y \beta_{xy}$ the total rate at which the walk exits from vertex $x$.

We define a *path* to be a sequence of vertices, denoted $\omega = (\omega_1, \ldots, \omega_n)$. We refer to $i = 1, ..., n$ as *time*. The probability $\mathbb{P}(\omega)$ that the random walk selects the path $\omega$ and then stops is

$$\mathbb{P}(\omega) = \frac{\lambda_{\omega_n}}{r_{\omega_n}} q(\omega), \tag{1}$$

$$q(\omega) = \frac{\beta_{\omega_1 \omega_2}}{r_{\omega_1}} \frac{\beta_{\omega_2 \omega_3}}{r_{\omega_2}} \cdots \frac{\beta_{\omega_{n-1} \omega_n}}{r_{\omega_{n-1}}}. \tag{2}$$

We call $q(\omega)$ the *weight function*. Note that this weight factorizes: If $\omega^{(a)} = (\omega_1^{(a)}, \ldots, \omega_n^{(a)})$ and $\omega^{(b)} = (\omega_1^{(b)}, \ldots, \omega_m^{(b)})$, with $\omega_n^{(a)} = \omega_1^{(b)}$, then the composition $\omega := \omega^{(a)} \circ \omega^{(b)} = (\omega_1^{(a)}, \ldots, \omega_n^{(a)} = \omega_1^{(b)}, \ldots, \omega_m^{(b)})$ of the paths $\omega^{(a)}$ and $\omega^{(b)}$ has weight $q(\omega) = q(\omega^{(a)}) q(\omega^{(b)})$.

One example to have in mind is

$$\mathcal{G} = \mathbb{Z}^d, \tag{3}$$

$$\lambda_x = \lambda, \tag{4}$$

$$\beta_{xy} = \beta_{yx} = 1_{x \sim y}. \tag{5}$$

This choice describes the simple symmetric nearest-neighbor random walk on $\mathbb{Z}^d$, stopped at a random time with rate $\lambda$; its length has an exponential distribution with mean $\lambda^{-1}$.

A second important example is a random walk in a finite box stopped on the boundary, described by the choice

$$\mathcal{G} = \{-l, \dots, l\}^d, \tag{6}$$

$$\lambda_x = \begin{cases} 0 & \text{if } x \in \{-l+1, \dots, l-1\}^d, \\ \infty & \text{otherwise,} \end{cases} \tag{7}$$

$$\beta_{xy} = \beta_{yx} = 1_{x \sim y}. \tag{8}$$

Given a path $\omega = (\omega_1, \dots, \omega_n)$, we define the *loop-erasure* procedure. The loop erasure is obtained by applying consecutively the *one-loop erasure*: look for the first time $i$ at which the path repeats a vertex, so $\omega_i = \omega_j$ for some $j < i$. The one-loop erasure of $\omega$ is the path $(\omega_1, \dots, \omega_j, \omega_{i+1}, \dots, \omega_n)$. We then apply the one-loop erasure to this new path, and continue until the path has no repeating vertices. The resulting path is the *loop erasure* of $\omega$, denoted LE($\omega$). It is self-avoiding, and when the initial path $\omega$ is a random walk, it is called the *loop-erased random walk* (LERW). If $\gamma$ is a LERW ending at $x$, the probability to generate it is

$$\mathcal{P}(\gamma) = \frac{\lambda_x}{r_x} \sum_{\omega : \text{LE}(\omega) = \gamma} q(\omega). \tag{9}$$

**Warning:** a *self-avoiding path* is a cominatorial object, i.e., it has no statistics. The loop-erased random walk is one possible distribution on the set of self-avoiding paths. It should not be confused with the *self-avoiding walk* or *self-avoiding polymer* commonly studied in the physics literature, which is another distribution on the set of self-avoiding paths. We distinguish the combinatorial object from the probabilistic one by using the term *path*.

## 3 Viennot's theorem

The main tool we use is a combinatorial theorem by Viennot [18]. It is part of the general theory of *heaps of pieces*. In our case it reduces to a relation between loop-erased random walks, and collections of loops. It allows us to calculate the total weight $\mathcal{P}(\gamma)$ of all paths $\omega$ whose loop erasure is $\gamma$, and represent the latter as a lattice model.

Before discussing the general case, consider a graph consisting of two vertices and one edge, and the following path,

$$\gamma = \;\text{\textcircled{$x$}}\!\longrightarrow\!\text{\textcircled{$y$}}. \tag{10}$$

The probability that such a path is the result of a loop-erased random walk leaving from $x$ and

arriving at $y$ is given by a geometric sum,

$$
\begin{aligned}
\mathcal{P}(\gamma) &= \;\; \underset{x}{\bigcirc}\!\!\longrightarrow\!\!\underset{y}{\bigcirc} + \underset{x}{\bigcirc}\!\!\!\!\!\!\Longrightarrow\!\!\!\!\!\!\underset{y}{\bigcirc} + \ldots \\
&= \frac{\lambda_y}{r_y}\left(\frac{\beta_{xy}}{r_x} + \frac{\beta_{xy}}{r_x}\frac{\beta_{yx}}{r_y}\frac{\beta_{xy}}{r_x} + \ldots\right) \\
&= \frac{\lambda_y}{r_y}q(\gamma)\frac{1}{1 - r_x^{-1}\beta_{xy}r_y^{-1}\beta_{yx}} \; .
\end{aligned}
\tag{11}
$$

We come back to this example later.

A *loop* is a path $\omega = (\omega_1, \ldots, \omega_{n-1}, \omega_n = \omega_1)$ where the first and last points are identical. We also require all vertices to be distinct (except $\omega_1$ and $\omega_n$), so it cannot be decomposed into smaller loops. Loops obtained from each other via cyclic permutations (dropping the repeated vertex $\omega_n$) are considered identical.

By a *collection of disjoint loops* we mean a set $L = \{C_1, C_2, \ldots\}$, each of whose elements is a loop, and the intersection of any pair of loops in $L$ is empty. We denote the *set of all such collections* by $\mathcal{L}$.

In order to formulate the theorem, we fix a self-avoiding path $\gamma$. We define the set $\mathcal{L}_\gamma$ to consist of the collections of disjoint loops in which no loop intersects $\gamma$. Then Viennot's theorem can be written as ($|L|$ being the number of loops)

$$
\mathcal{A}(\gamma) := q(\gamma)\sum_{L\in\mathcal{L}_\gamma}(-1)^{|L|}\prod_{C\in L}q(C) = \sum_{\omega:\text{LE}(\omega)=\gamma}q(\omega) \times \sum_{L\in\mathcal{L}}(-1)^{|L|}\prod_{C\in L}q(C).
\tag{12}
$$

On the l.h.s. one sums over the ensemble of collections of loops which do not intersect $\gamma$, giving each collection a weight $(-1)^{|L|}\prod_{C\in L}q(C)$. We later calculate this object using field-theory. The r.h.s. contains two factors. The first is the weight to find the LERW path $\gamma$, our object of interest. The second is the partition function

$$
\mathcal{Z} := \sum_{L\in\mathcal{L}}(-1)^{|L|}\prod_{C\in L}q(C)
\tag{13}
$$

of the loop model on the left-hand side. Assuming the walk to stop at $x$, this relation can be read as

$$
\mathcal{P}(\gamma) = \frac{\lambda_x}{r_x}\sum_{\omega:\text{LE}(\omega)=\gamma}q(\omega) = \frac{\lambda_x}{r_x}\frac{q(\gamma)\sum_{L\in\mathcal{L}_\gamma}(-1)^{|L|}\prod_{C\in L}q(C)}{\mathcal{Z}}.
\tag{14}
$$

Let us verify this formula in the elementary example of equation (10). On the l.h.s, since all loops intersect $\gamma$, the only collection of disjoint loops in $L_\gamma$ is the empty collection that contains no loops. Its size $|L| = 0$, so the sign is 1, and since the empty product equals 1 we obtain $q(\gamma)$. On the right-hand side, the first term $\sum_{\omega:\text{LE}(\omega)=\gamma}q(\omega)$ is the sum that we have calculated, $q(\gamma)\frac{1}{1-r_x^{-1}\beta_{xy}r_y^{-1}\beta_{yx}}$. The second term is the sum over two possible collections: the empty collection, and the one that contains the single loop $L = \{(x, y, x)\}$. The empty collection gives weight 1, and the single-loop collection has weight $\frac{\beta_{xy}}{r_x}\frac{\beta_{yx}}{r_y}$, with a minus sign from $(-1)^{|L|}$. Putting everything together,

$$
q(\gamma) = q(\gamma)\frac{1}{1-r_x^{-1}\beta_{xy}r_y^{-1}\beta_{yx}} \times \left(1 - \frac{\beta_{xy}}{r_x}\frac{\beta_{yx}}{r_y}\right),
\tag{15}
$$

which indeed confirms Viennot's theorem in this specific case.

The idea for the proof of equation (12) is to consider a pair $\{\omega, L\}$ constructed as follows: Take a path $\omega$ such that $\text{LE}(\omega) = \gamma$ and an *arbitrary* collection $L$ of disjoint loops. Our goal is

to construct another pair $\{\omega', L'\}$ by transferring a loop from $L$ to $\omega$ or vice versa, depending on where the loop originally was. For example,

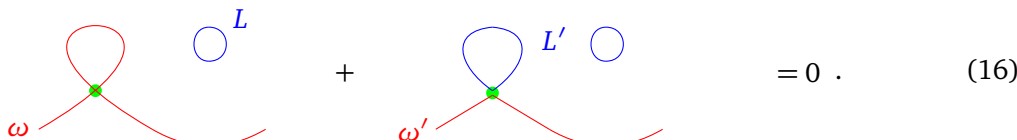

$$+ \qquad = 0 \; . \qquad (16)$$

In the first drawing, the left loop is part of $\omega$, whereas in the second one it is part of $L'$. These terms cancel, as $(-1)^{|L|} = -(-1)^{|L'|}$, and all other factors are identical. After each such pair is canceled, we are left with the terms in which it is impossible to transfer a loop from $\omega$ to $L$ or vice versa. These are exactly the terms in the l.h.s of equation (12).

For this procedure to work we need to make sure that we cannot obtain the same pair $\{\omega', L'\}$ starting form two different pairs $\{\omega, L\}$. In order to achieve this, we use the following prescription. Start walking along $\omega$, until

1. we reach a vertex $\omega_i$ that belongs to some $C = (\omega_i = c_1, c_2 \ldots, c_m = \omega_i) \in L$, or

2. we reach a vertex $\omega_i$ that does not belong to any $C$, but that we have already seen before, i.e., $\omega_j = \omega_i$ for $j < i$.

In the first case, we transfer $C$ to $\omega$, i.e.,

$$\omega' = (\omega_1, \ldots, \omega_i, c_2, \ldots, c_{m-1}, \omega_i, \ldots, \omega_n), \text{ and } L' = L \setminus \{C\}. \qquad (17)$$

In the second case, we apply the one-loop erasure to $\omega$, and transfer the erased loop to $L$,

$$\omega' = (\omega_1, \ldots, \omega_j, \omega_{i+1}, \ldots, \omega_n), \text{ and } L' = L \cup \{(\omega_j, \omega_{j+1}, \ldots, \omega_i = \omega_j)\}. \qquad (18)$$

Note that disjointness of the loop collections is preserved under the transfer, and that the loop erasure of $\omega'$ remains $\gamma$.

Let us consider some special cases: the first example is $L$ being the empty set. If $\omega$ contains no loops, then $\omega = \gamma$. This term appears in the l.h.s. of equation (12). If, on the other hand, $\omega$ does contain a loop, we take $C = (\omega_j, \ldots, \omega_i)$ to be the first loop in the loop erasure of $\omega$. Then $\omega' = (\omega_1, \ldots, \omega_j, \omega_{i+1}, \ldots, \omega_n)$ and $L' = C$. The same thing happens when $L$ is not empty, but none of the loops in $L$ intersects $\omega$. Loops in $L$ that do not intersect $\omega$ act as spectators, as in equation (16).

A second example, is when $\omega$ winds $k$ times around a loop $C$ that also belongs to $L$. If $C$ is the first loop that we encounter, then $L' = L \setminus \{C\}$, and $\omega'$ has the same trace as $\omega$, but it winds $k + 1$ times around $C$ rather than $k$ times.

Third, if a loop $C \in L$ intersects $\omega$ in more than one point, the instructions above determine how this loop is attached. For example, if

$$\{\omega, L\} = \quad \longrightarrow \bigcirc \longrightarrow \quad , \qquad (19)$$

then

$$\omega' = \quad \longrightarrow \bigcirc \longrightarrow \quad \neq \quad \longrightarrow \bigcirc \longrightarrow \quad . \qquad (20)$$

# 4  A lattice action with two complex fermions and one complex boson

Our goal is to write a lattice action which generates the l.h.s. of equation (12), namely

$$\mathcal{A}(\gamma) = q(\gamma) \sum_{L \in \mathcal{L}_\gamma} (-1)^{|L|} \prod_{C \in L} q(C). \tag{21}$$

This can be achieved with an action based on one pair $(\phi, \phi^*)$ of complex conjugate fermionic fields. While this theory sums over all paths $\gamma$, yielding back the random-walk propagator, it contains no information on the erasure. In order to answer whether the resulting loop-erased path passes through a given point $y$ it is necessary to use more fields. The simplest such setting consists of two pairs of complex conjugate fermionic fields $(\phi_1, \phi_1^*)$, and $(\phi_2, \phi_2^*)$, as well as a pair of complex conjugate bosonic fields $(\phi_3, \phi_3^*)$. When appearing in a loop, the latter cancels one of the fermions.

In the last section we saw that the loop-erased random walk is tightly connected to the combinatorics of non-intersecting loops and paths. When trying to express such objects using a field theory, the non-intersection property appears naturally with fermionic fields, but not with bosonic ones. In the theory that we construct we need to have both fermionic and bosonic components. To ensure non-intersection also for bosonic fields, we introduce the *nilpotent bosonic field* in analogy to the Grassmannian fermionic field. Denoting $g(\phi)$ the *grade* of a field $\phi$, we set $g(\phi) = 1$ for fermionic fields and $g(\phi) = 0$ for bosonic fields,

$$g(\phi_1) = g(\phi_1^*) = g(\phi_2) = g(\phi_2^*) = 1, \tag{22}$$
$$g(\phi_3) = g(\phi_3^*) = 0. \tag{23}$$

The fields are employed with *graded commutation relations*, so that for all $x$ and $y$

$$\phi(x)\psi(y) = (-1)^{g(\phi)g(\psi)}\psi(y)\phi(x), \tag{24}$$
$$\phi(x)^2 = 0. \tag{25}$$

The second relation is a consequence of the first one for fermions, but an additional rule for bosons.

The *path integral* for both fermions and bosons is defined as the Berezin integral [19],

$$\int_{\phi^*(x)} \int_{\phi(x)} \mathcal{F}(\phi, \phi^*) := \frac{d}{d\phi^*(x)} \frac{d}{d\phi(x)} \mathcal{F}(\phi, \phi^*)\Big|_{\phi(x)=\phi^*(x)=0}. \tag{26}$$

This implies a correlation 1 for $\phi_i(x)\phi_i^*(x)$, in this order.

One way to represent a nilpotent boson is to write it as a product of two fermions [11]. In this representation, using the Berezin integral for these fermions yields equation (26) for the nilpotent bosons, making it applicable for all fields. As it is not possible to integrate out these fermionic fields in exchange for a functional determinant, as is done e.g. to treat superconductivity [20], the intuition behind this construction is misleading, and we will not pursue it any further.

We define our action as

$$\phi^*(y)\phi(x) := \sum_{i=1}^{3} \phi_i^*(y)\phi_i(x), \tag{27}$$

$$e^{-\mathcal{S}} = \prod_x \left\{ \left[1 + \sum_y \beta_{xy} \phi^*(y)\phi(x)\right] \prod_{i=1}^{3} \left[1 + r_x \phi_i(x)\phi_i^*(x)\right] \right\}. \tag{28}$$

We can calculate the partition function $\mathcal{Z} = \int e^{-\mathcal{S}}$ by expanding the products. Each term of the expansion is represented as a colored multigraph: whenever we encounter a term $\beta_{xy}\phi_i^*(y)\phi_i(x)$, we place an edge from $x$ to $y$ of color $i$; whenever we encounter a term $r_x\phi_i(x)\phi_i^*(x)$ we place a self-loop of color $i$ at $x$. By *self-loop at $x$* we mean an edge from $x$ to $x$. This representation is one-to-one, since knowing the colored multigraph $G$, we can recover the corresponding term in the expansion by taking $\beta_{xy}\phi_i^*(y)\phi_i(x)$ for each colored edge and $r_x\phi_i(x)\phi_i^*(x)$ for each self-loop.

The multigraphs that contribute to $\mathcal{Z}$ must have *complete degrees* (i.e., each vertex has incoming and outgoing degree 1 of each color), and due to the structure of the action each vertex can have at most one outgoing edge, not counting self-loops. By taking outside the self-loops, the graphs left are collections of disjoint colored loops, with contribution

$$\mathcal{Z}_0 (-1)^{\#\text{fermionic loops}} \prod_{C \text{ loop of } G} q(C). \tag{29}$$

$$\mathcal{Z}_0 := \int_{\phi^*,\phi} \prod_x \prod_{i=1}^{3} \left[ 1 + r_x \phi_i(x)\phi_i^*(x) \right] = \prod_x r_x^3. \tag{30}$$

The reason for the factor of $(-1)^{\#\text{fermionic loops}}$ is that Berezin integration requires the fields to come in a certain order. As bosons commute, we may reorder them as we wish; reordering fermions may cost a sign. The self-loops appear in the right order, $\phi_i(x)\phi_i^*(x)$.

In contrast, if the graph contains a fermionic loop $(x_1,\ldots,x_n = x_1)$, then the corresponding term in the expansion has edges $(x_{n-1},x_n),(x_{n-2},x_{n-1}),\ldots,(x_1,x_2)$, and is given by $\beta_{x_{n-1}x_n}\phi_i^*(x_n)\phi_i(x_{n-1})\beta_{x_{n-2}x_{n-1}}\phi_i^*(x_{n-1})\phi_i(x_{n-2})\ldots\beta_{x_1x_2}\phi_i^*(x_2)\phi_i(x_1)$. In order to apply Berezin integration, we need to move the last variable $\phi_i(x_1)$ to the beginning. This requires an odd number of exchanges, hence the minus sign.

Summing equation (29) over all possible colorings and all graphs, we obtain

$$\mathcal{Z} = \mathcal{Z}_0 \sum_{L \in \mathcal{L}} (-1)^{|L|} \prod_{C \in L} q(C). \tag{31}$$

This is, up to the prefactor of $\mathcal{Z}_0$, the partition function defined in equation (13).

In order to assess whether a point $b$ belongs to a loop-erased random walk from $a$ to $c$ after erasure, we fix the three vertices $a, b$ and $c$ and consider the observable

$$U(a,b,c) = \lambda_c r_b \left\langle \phi_2(c)\phi_2^*(b)\phi_1(b)\phi_1^*(a) \right\rangle. \tag{32}$$

The graphs that contribute to the integral consist of a self-avoiding path $\gamma$ and a collection $L$ of disjoint self-avoiding colored loops such that (see Figure 1):

1. $\gamma$ is a path from $a$ to $c$ passing through $b$. The edges of $\gamma$ between $a$ and $b$ have color 1, and the edges between $b$ and $c$ have color 2.

2. Fix $C \in L$. If the color of $C$ is 2 then it cannot intersect $\gamma$. If its color is 1 or 3, it can only intersect $\gamma$ at the point $c$.

In the latter case, the contribution to $\int \phi_2(c)\phi_2^*(b)\phi_1(b)\phi_1^*(a)e^{-\mathcal{S}}$ is

$$\mathcal{Z}_0 (-1)^{\#\text{fermionic loops}} r_b^{-1} r_c^{-1} q(\gamma) \prod_{C \in L} q(C). \tag{33}$$

We now sum over all possible colorings of the loops. Since loops that intersect $c$ may have either color 1 or 3, one fermionic and one bosonic, they cancel, leaving only graphs in which

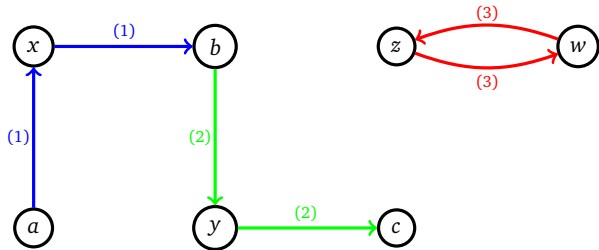

Figure 1: An example of a diagram that contributes to $U$. Our coloring conventions are blue for $(\phi_1, \phi_1^*)$, green for $(\phi_2, \phi_2^*)$, and red for $(\phi_3, \phi_3^*)$.

loops do not intersect $\gamma$ at all. The other loops, as before, give a factor of $-1$. We have therefore established that

$$\int_{\phi^*,\phi} \phi_2(c)\phi_2^*(b)\phi_1(b)\phi_1^*(a)e^{-\mathcal{S}} = \mathcal{Z}_0\, r_b^{-1} r_c^{-1} \sum_{\substack{\gamma:a\to b\to c \\ \text{self-avoiding}}} q(\gamma) \sum_{L\in\mathcal{L}_\gamma} (-1)^{|L|} \prod_{C\in L} q(C), \quad (34)$$

where the sum is over all self-avoiding paths from $a$ to $c$ passing through $b$.

What remains to be done is to combine Viennot's theorem (12) with equations (31) and (34),

$$\sum_{\substack{\gamma:a\to c \\ b\in\gamma}} \mathcal{P}(\gamma) \equiv \sum_{\substack{\omega:a\to c \\ b\in\text{LE}(\omega)}} \mathbb{P}(\omega) = \sum_{\substack{\gamma:a\to b\to c \\ \text{self-avoiding}}} \sum_{\omega:\text{LE}(\omega)=\gamma} q(\omega)\frac{\lambda_c}{r_c}$$

$$= \frac{\lambda_c}{r_c} \sum_{\substack{\gamma:a\to b\to c \\ \text{self-avoiding}}} q(\gamma) \frac{\sum_{L\in\mathcal{L}_\gamma}(-1)^{|L|}\prod_{C\in L}q(C)}{\sum_{L\in\mathcal{L}}(-1)^{|L|}\prod_{C\in L}q(C)}$$

$$= U(a,b,c). \quad (35)$$

Note that the above formalism allows us to consider LERWs starting at $a$, going through $b_1$ and $b_2$ in this order, and ending at $c$,

$$U(a,b_1,b_2,c) = \lambda_c r_{b_1} r_{b_2} \left\langle \phi_3(c)\phi_3^*(b_2)\phi_2(b_2)\phi_2^*(b_1)\phi_1(b_1)\phi_1^*(a) \right\rangle. \quad (36)$$

For more intermediate points, one needs more fields: Each addition of one complex conjugate pair of bosonic fields together with one complex conjugate pair of fermionic fields allows us to have two more intermediate points. We will not pursue this direction any further.

Before concluding this section, let us rewrite the action $\mathcal{S}$ explicitly. The starting point is equation (28),

$$e^{-\mathcal{S}} = \prod_x \left\{ \left[1 + \sum_y \beta_{xy}\phi^*(y)\phi(x)\right]\prod_{i=1}^3 \left[1 + r_x\phi_i(x)\phi_i^*(x)\right] \right\}. \quad (37)$$

We need to take the logarithm of this equation. Due to the nilpotence of the fields, the expansions stop at the latest at order three, the number of colors,

$$\ln\left(1 + r_x\phi_i(x)\phi_i^*(x)\right) = r_x\phi_i(x)\phi_i^*(x),$$

$$\ln\left(1 + \sum_y \beta_{xy}\phi^*(y)\phi(x)\right) = \sum_y \beta_{xy}\phi^*(y)\phi(x) - \frac{1}{2}\left[\sum_y \beta_{xy}\phi^*(y)\phi(x)\right]^2$$
$$+ \frac{1}{3}\left[\sum_y \beta_{xy}\phi^*(y)\phi(x)\right]^3. \quad (38)$$

Putting the fields in the "natural order" ($\phi_i^*$ before $\phi_i$) yields

$$
\begin{aligned}
\mathcal{S} = \sum_x \Big\{ &r_x \big[ \phi_1^*(x)\phi_1(x) + \phi_2^*(x)\phi_2(x) - \phi_3^*(x)\phi_3(x) \big] - \sum_y \beta_{xy}\phi^*(y)\phi(x) \\
&+ \frac{1}{2}\Big[ \sum_y \beta_{xy}\phi^*(y)\phi(x) \Big]^2 - \frac{1}{3}\Big[ \sum_y \beta_{xy}\phi^*(y)\phi(x) \Big]^3 \Big\} \\
= \sum_x \Big\{ &r_x\big[ \phi_1^*(x)\phi_1(x) + \phi_2^*(x)\phi_2(x) - \phi_3^*(x)\phi_3(x) \big] - \sum_y \beta_{xy}\big[ \phi^*(y) - \phi^*(x) \big]\phi(x) \\
&- \Big( \sum_y \beta_{xy} \Big)\phi^*(x)\phi(x) + \frac{1}{2}\Big[ \sum_y \beta_{xy}\phi^*(y)\phi(x) \Big]^2 - \frac{1}{3}\Big[ \sum_y \beta_{xy}\phi^*(y)\phi(x) \Big]^3 \Big\} \\
= \sum_x \Big\{ &m_x^2\phi_1^*(x)\phi_1(x) + m_x^2\phi_2^*(x)\phi_2(x) + \mu_x^2\phi_3^*(x)\phi_3(x) - \nabla_\beta^2\phi^*(x)\phi(x) \\
&+ \frac{1}{2}\Big[ \sum_y \beta_{xy}\phi^*(y)\phi(x) \Big]^2 - \frac{1}{3}\Big[ \sum_y \beta_{xy}\phi^*(y)\phi(x) \Big]^3 \Big\},
\end{aligned}
\tag{39}
$$

with

$$
m_x^2 = r_x - \sum_y \beta_{xy}, \qquad \mu_x^2 = -r_x - \sum_y \beta_{xy}.
\tag{40}
$$

This theory contains two fermions with mass $m_x^2 = \lambda_x$ and a nilpotent boson with mass $\mu_x^2 = -\lambda_x - 2\sum_y \beta_{xy}$. In this formulation, the cancelation between the nilpotent boson and one of the fermions is not obvious.

## 5 Trading nilpotent bosons for standard bosons

The purpose of this section is to show that nilpotent bosons behave like interacting standard bosons, and to rewrite the action with the latter. To simplify our considerations, we do not write the fermions; including them later is straightforward.

The bosonic part of equation (39) is

$$
\mathcal{S}^{(3)} = \sum_x \Big[ \mu_x^2\phi_3^*(x)\phi_3(x) - \nabla_\beta^2\phi_3^*(x)\,\phi_3(x) \Big].
\tag{41}
$$

As in the previous section, we can expand the corresponding partition function, obtaining

$$
\mathcal{Z}^{(3)} = \mathcal{Z}_0^{(3)} \sum_{L \in \mathcal{L}} \prod_{C \in L} q(C),
\tag{42}
$$

where $\mathcal{Z}_0^{(3)} = \prod_x r_x$.

We now consider an interacting bosonic field theory with a standard pair of complex conjugate bosonic fields $\{\chi(x)\}_{x \in \mathcal{G}}$, and action

$$
\mathcal{S}' = \sum_x \Big[ r_x\chi^*(x)\chi(x) - \log\Big( 1 + \sum_y \beta_{xy}\chi^*(y)\chi(x) \Big) \Big].
\tag{43}
$$

The prime denotes objects evaluated with a standard pair of complex conjugate bosonic fields, and this action. Consider the corresponding partition function

$$
\mathcal{Z}' = \int_{\chi^*,\chi} e^{-\mathcal{S}'} = \int_{\chi^*,\chi} e^{-\sum_x r_x\chi^*(x)\chi(x)} \prod_x \Big( 1 + \sum_y \beta_{xy}\chi^*(y)\chi(x) \Big).
\tag{44}
$$

This is a Gaussian integral, and we calculate it by expanding the product over $x$. As before, we represent terms in the product as a graph, placing an edge from $x$ to $y$ whenever the term $\beta_{xy}\chi^*(y)\chi(x)$ appears. Unlike in the nilpotent case, we have no self-loops in the graph. Moreover, since the directed edge from $x$ to $y$ appears in only one factor of the product, there are no double edges, yielding a graph rather than a multigraph. One more constraint, coming from the form of the action, is that each vertex of this graph has out-degree at most 1.

The diagrams that contribute to $\mathcal{Z}'$ are those in which at each vertex $x$ the fields appear as $(\chi^*(x)\chi(x))^n$ for some $n$. Since the outgoing degree of $x$ is at most 1, the power $n$ is at most 1. Thus each vertex has incoming and outgoing degrees that are either both 1 or both 0. We obtain

$$\mathcal{Z}' = \mathcal{Z}'_0 \sum_{L\in\mathcal{L}}\prod_{C\in L} q(C), \tag{45}$$

$$\mathcal{Z}'_0 = \prod_x \frac{\pi}{r_x}. \tag{46}$$

We see that the diagrams contributing to the (standard) bosonic theory (43) are the same as those contributing to the nilpotent bosonic field. Care has to be taken when considering correlation functions, since external fields may have a slightly different behavior. This can be done in general, but since the observable $U(a,b,c)$ defined in equation (36) has only fermionic external fields, we can discard these details.

In order to better understand the action defined in equation (43), let us write it explicitly:

$$\begin{aligned}
\mathcal{S}' &= \sum_x\left\{ r_x\chi^*(x)\chi(x) - \sum_y \beta_{xy}\chi^*(y)\chi(x) + \sum_{k=2}^\infty \frac{(-1)^k}{k}\Big[\sum_y \beta_{xy}\chi^*(y)\chi(x)\Big]^k\right\}\\
&= \sum_x\left\{ m_x^2\chi^*(x)\chi(x) - \nabla_\beta^2\chi^*(x)\chi(x) + \sum_{k=2}^\infty \frac{(-1)^k}{k}\Big[\Big(\nabla_\beta^2 + \sum_y \beta_{xy}\Big)\chi^*(x)\chi(x)\Big]^k\right\}. \quad (47)
\end{aligned}$$

We see from this equation that the nilpotent boson with mass $\mu$ is equivalent to an interacting boson with mass $m$. That is, the action of our $\mathcal{O}(-2)$ field theory given in equation (39) describes a field with two fermionic components and one bosonic one, all with the same mass.

The identities laid out above are exact, and hold for any graph and any set of parameters. If we consider a lattice in dimension $d > 2$, after dropping terms that are irrelevant in the scaling limit, only the term $k = 2$ survives. Further dropping the Laplacian in this term, we are left with a standard $\phi^4$ theory (see, e.g., [21]).

To conclude this section, we return to the loop-erased random walk, but this time with a field $\phi$, containing two complex fermions and one complex (*not* nilpotent) boson. The *exact* action for the loop-erased random walk is

$$\mathcal{S}' = \sum_x\left\{ m_x^2\phi^*(x)\phi(x) - \nabla_\beta^2\phi^*(x)\phi(x) + \sum_{k=2}^\infty \frac{(-1)^k}{k}\Big[\Big(\nabla_\beta^2 + \sum_y \beta_{xy}\Big)\phi^*(x)\phi(x)\Big]^k\right\}. \tag{48}$$

We define $U'$ in the same manner as in equation (36),

$$U'(a,b,c) = \lambda_c r_b \left\langle \phi'_2(c)\phi'^*_2(b)\phi'_1(b)\phi'^*_1(a)\right\rangle, \tag{49}$$

where $\langle\cdot\rangle$ is understood with respect to $\mathcal{S}'$. Then equation (35) holds, and

$$\sum_{\substack{\gamma:a\to c\\ b\in\gamma}} \mathcal{P}(\gamma) \equiv \sum_{\substack{\omega:a\to c\\ b\in\mathrm{LE}(\omega)}} \mathbb{P}(\omega) = U'(a,b,c). \tag{50}$$

Again, in $d > 2$, the scaling limit of this theory is the $\phi^4$ theory. Perturbative calculations indicate that the theory remains valid in dimension $d = 2$ [22].

# 6 Selfavoiding polymers, and other applications

In the considerations above, we used two fermionic fields, and one bosonic one. The latter could be chosen either as a standard bosonic field, or as nilpotent. We saw that both formulations are exact. There are other physical systems which have the same lattice expansion, albeit with a different number $N_{\text{b}}$ of bosonic and $N_{\text{f}}$ fermionic fields:

(i) one bosonic $N_{\text{b}} = 1$ and one fermionic field $N_{\text{f}} = 1$: Then all loops cancel, and each self-avoiding path $\gamma$ appears with a weight one. These are self-avoiding polymers. The action (48), or its resummed version, are exact. Keeping only the leading term $k = 2$, and dropping in the latter the Laplacian, we arrive at weakly self-avoiding walks, which are in the same universality class. This was first observed in 1972 by de Gennes [23], and later elaborated by many authors, see [21, 24–28] and references therein. Similar formulas to ours appear in the work by Brydges, Imbrie and Slade [29].

(ii) the standard lattice $\mathcal{O}(n)$ model, defined by its loop representation which allows one to take $n$ arbitrary, corresponds to $n = 2(N_{\text{b}} - N_{\text{f}})$. In the graph representation, the factor of 2 is due to the two possible choices of the direction in a loop. In the field theory $n$ counts the number of real components, and a complex field has both a real and an imaginary part. See e.g. [30].

Using equation (48) with one real field yields the exact action for the Ising model on a 3-regular graph such as the honeycomb lattice. It allows one to evaluate lattice observables in the field theory exactly. Dropping irrelevant terms leads to $\phi^4$ theory, without the need to employ the standard [21, 31, 32] coarse-graining procedure.

# 7 Conclusion and further questions

We showed how LERW observables can be expressed as correlation functions of a field theory with two complex fermions, and one complex boson. This mapping is an exact combinatorial identity for a field theory with specific interactions. It explains the result in [9,10], by observing that, at least for $d > 2$, the large-scale limit of this field theory is the standard $\phi^4$ theory. Our approach has the advantage to give exact equalities on all graphs.

One convenient way to prove this identity was by introducing a field of *nilpotent* bosons. While commuting, they are as fermions not allowed to intersect, and may be interpreted as *hard-core bosons*, popular in hard condensed matter physics [20]. A nilpotent boson is equivalent to an interacting bosonic field; the non-intersection property simplifies the combinatorial analysis. It suggests a simplifying alternative for the action of the $\mathcal{O}(n)$ loop-model [6, 7] which in those works is only exact on 3-regular graphs such as the honeycomb lattice.

The results and methods laid out here lead to various openings: An intriguing problem is the search for a combinatorial object corresponding to the $\mathcal{O}(-1)$ theory. Exploring more properties of nilpotent bosons, and studying how they behave in different theories, is another interesting direction. Finally, the transition rates $\beta_{xy}$ are general. This allows one to study situations with drift, gauge fields, or disorder.

## Acknowledgements

We thank Andrei Fedorenko and Tyler Helmuth for valuable discussions, and a careful reading of the manuscript.

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
