# Peer review of "An exact mapping between loop-erased random walks and an interacting field theory with two fermions and one boson"

_SciPost Physics, doi:SciPost Phys. 9, 063 (2020)_

## Round 1 · Referee Report · Ilya Gruzberg (Referee 1) · 2020-8-4

Strengths

  1. The paper contains a new exact relation between a probabilistic object, which is intrinsically non-local (loop erased random walk), and a local field theory. The relation holds for any directed graph.
  2. The paper is well written and organized, clearly stating its results and outlining the derivations.
  3. The paper is sufficiently compact to be read reasonably quickly in order to grasp the main results.
  4. The paper provide enough details to reproduce all the intermediate steps.

Report

I liked reading and refereeing the paper, which is written in a very logical way, and includes sufficient amount of detail for readers to either understand the authors' point right away or to be able to reproduce the necessary steps for themselves.
The paper presents a new and exact relation between loop-erased walks on arbitrary directed graphs and discrete field theories of two complex fermions and one complex boson. Such a relation was known before from the work of one of the authors (and collaborators), where the complex boson involved in the construction was nilpotent. The present paper demonstrates that the nilpotetn boson can be replaced a much more familiar canonical complex bosonic field. The applications and extensions of this work are numerous and partially summarized by the authors at the end of the paper.

Requested changes

  1. I have never before encountered nilpotent bosons, and would like the authors to provide some clarifications. I had to go to the previously published paper by T. Helmuth and A. Shapira to make sense of the nilpotent bosons. They turned out to be even members of a Grassman algebra made up by to fermionc fields. However, this is not sufficient to understand why the definition of the functional integral for the nilpotent bosons can be taken as the Berezin integral, literally in parallel with the fermionic degrees of freedom. Therefore, I request that the authors provide an explanation of this choice of the functional integral, and its consequences: what is the result for a Gaussian integral of this form, and what is the statement of the Wick's theorem for the nilpotent bosons?

---

## Round 1 · Referee Report · Anonymous (Referee 2) · 2020-9-18

Strengths

The manuscript presents a new way of establishing field theory formulations of certain statistical models on lattices, which has the advantage that the first step on the lattice model is an exact combinatorial one and does not rely on coarse-graining.

The combinatorics of the method is interesting in its own right.

The new combinatorial approach is clearly compared with prior field theory literature on the statistical models.

The manuscript is very well written.

Weaknesses

The manuscript does not state the extent to which questions about the loop-erased random walk can be addressed in the field theory. Besides partition functions, they give one interesting and non-trivial example: the probability that a loop-erased random walk goes via one given vertex. Does the field theory allow, e.g., for the calculation of the probability to go via several given vertices?

Report

In this manuscript, a field theory formulation is developed for the stochastic model of loop-erased random walks on a lattice. The key insight is to use a nilpotent boson field, together with two fermion fields, to simplify the arguments on the lattice. The authors also reformulate the field theory in such a way that the nilpotent boson is replaced by an ordinary boson. The results are correct and combinatorially interesting, and they can be used to obtain notable simplifications to some field theory formulations of statistical models by making the first reformulation as a lattice field theory and exact combinatorial statement that does rely on coarse-graining. The manuscript is generally well-written, but I would ask the authors to consider the specific comments below. Once these have been addressed, I am happy to recommend this clear and interesting article for publication in SciPost Physics.

Requested changes

1) • p. 3, Example (10): Clarify that the graph is assumed to contain no other directed edges except between x and y (or perhaps a milder but similar assumption). Perhaps the authors in fact thought of the graph consisting only of these two vertices, which should then be stated clearly.

2) • p. 3, Definition of loop: The text defines loops as paths whose first and last vertices are equal and all other vertices are distinct. For the following combinatorics to work out correctly, this definition should be modified so that any two paths related by a cyclic permutation of vertices are considered the same loop. A possible terminology for this is (oriented) unrooted loops, as opposed to rooted loops.

3) • p. 3, Definition of $\mathcal{L}_{\gamma}$: Sentence: “We define the set $\mathcal{L}_{\gamma}$ to be the collections of…” -> “We define $\mathcal{L}_{\gamma}$ to be the set of collections of…” or “We define the set $\mathcal{L}_{\gamma}$ to consist of the collections of…”.

4) • p. 4, Paragraph of Eqn. (17)-(18) or of Eqn. (16): It is worth emphasizing that in the prescription, whenever a loop is transferred from the walk to the collection of loops, the disjointness of the loop collection is indeed preserved.

5) • p. 5, Paragraph of Eqn. (21): The terminology is otherwise accurate in referring to “pairs of complex conjugate fields”, except at the first mention of them. Change: “one pair $(\phi,\phi^{*})$ of complex fermionic fields” -> “one pair $(\phi,\phi^{*})$ of complex conjugate fermionic fields”.

6) • p. 6, Definition of “graded commutation relations”: The intended meaning of Eqn. (24) and (25) can probably be guessed correctly, but its current statement is not sufficiently clear. Specifically, (24) should be imposed for $\phi(x)\psi(y)$ for any $x,y$, whereas (25) should only be imposed for $\phi(x)\phi(y)$ when $x=y$.

7) • p. 6, Eqn (28): One further pair of parentheses would avoid any potential misunderstanding about whether the product $\prod_{i}$ is inside the product $\prod_{x}$ (as is should be), or whether the two are separate.

8) • p. 7, Paragraph of Eqn (30): The most interesting observable that the authors calculate in the field theory is $U(a,b,c)$, proportional to the probability that the loop-erased random walk from $a$ to $c$ goes via $b$. It appears very natural to ask about a generalization: does the field theory at least in principle also allow for the calculation of the probability that the loop-erased random walk from $a$ to $c$ goes via all of $b_{1},\ldots,b_{n}$? If the answer is no, the field theory formulation is still quite interesting, but it seems appropriate to briefly admit that there are questions about loop-erased random walks that can not be addressed in the theory. If the answer is yes, I recommend mentioning this interesting generalization and giving a brief hint for how it is done.

9) • p. 9, Paragraph of Eqn. (42): “standard bosonic fields” -> “standard pair of complex conjugate bosonic fields”.

10) • p. 11, Last paragraph before Section 7: “on a 3-regular graph as the honeycomb lattice” -> “on a 3-regular graph such as the honeycomb lattice”.

---

## Round 2 · Referee Report · Ilya Gruzberg (Referee 1) · 2020-10-20

Report

I am satisfied with the response of the authors to referees. I believe the paper can be published in tis present form

---

## Round 2 · Referee Report · Anonymous (Referee 2) · 2020-10-20

Report

The authors have addressed all the issues in my earlier report appropriately. I recommend this interesting article for publication in SciPost Physics.

---

## Round 2 · Author Response

We would first like to thank both referees for the detailed reviews.

The points raised in the second report concerning specific corrections were addressed as suggested by the referee.

As requested in the first report, we added details on the interpretation of the nilpotent bosons as even members of the Grassmann algebra and how this relates to the Berezin integral. The discussion on the physical and combinatorial meaning of these nilpotent bosons appears in section 5 and is summarized in section 7.

---

## Round 2 · List of Changes

1) • p. 3, Example (10): Clarify that the graph is assumed to contain no other directed edges except between x and y (or perhaps a milder but similar assumption). Perhaps the authors in fact thought of the graph consisting only of these two vertices, which should then be stated clearly.

Response: Clarification added.

2) • p. 3, Definition of loop: The text defines loops as paths whose first and last vertices are equal and all other vertices are distinct. For the following combinatorics to work out correctly, this definition should be modified so that any two paths related by a cyclic permutation of vertices are considered the same loop. A possible terminology for this is (oriented) unrooted loops, as opposed to rooted loops.

Response: The definition was modified

3) • p. 3, Definition of Lγ: Sentence: “We define the set Lγ to be the collections of…” -> “We define Lγ to be the set of collections of…” or “We define the set Lγ to consist of the collections of…”.

Response: Changed as suggested by the referee.

4) • p. 4, Paragraph of Eqn. (17)-(18) or of Eqn. (16): It is worth emphasizing that in the prescription, whenever a loop is transferred from the walk to the collection of loops, the disjointness of the loop collection is indeed preserved.

Response: Clarification was added.

5) • p. 5, Paragraph of Eqn. (21): The terminology is otherwise accurate in referring to “pairs of complex conjugate fields”, except at the first mention of them. Change: “one pair (ϕ,ϕ∗) of complex fermionic fields” -> “one pair (ϕ,ϕ∗) of complex conjugate fermionic fields”.

Response: Fixed according to the referee's suggestion.

6) • p. 6, Definition of “graded commutation relations”: The intended meaning of Eqn. (24) and (25) can probably be guessed correctly, but its current statement is not sufficiently clear. Specifically, (24) should be imposed for ϕ(x)ψ(y) for any x,y, whereas (25) should only be imposed for ϕ(x)ϕ(y) when x=y.

Response: We rewrote the relations more explicitly.

7) • p. 6, Eqn (28): One further pair of parentheses would avoid any potential misunderstanding about whether the product ∏i is inside the product ∏x (as is should be), or whether the two are separate.

Response: Added parentheses.

8) • p. 7, Paragraph of Eqn (30): The most interesting observable that the authors calculate in the field theory is U(a,b,c), proportional to the probability that the loop-erased random walk from a to c goes via b. It appears very natural to ask about a generalization: does the field theory at least in principle also allow for the calculation of the probability that the loop-erased random walk from a to c goes via all of b1,…,bn? If the answer is no, the field theory formulation is still quite interesting, but it seems appropriate to briefly admit that there are questions about loop-erased random walks that can not be addressed in the theory. If the answer is yes, I recommend mentioning this interesting generalization and giving a brief hint for how it is done.

Response: We added a paragraph after equation (35), explaining how our technique could be adapted in order to find higher-order observables.

9) • p. 9, Paragraph of Eqn. (42): “standard bosonic fields” -> “standard pair of complex conjugate bosonic fields”.

Response: Fixed as suggested by the referee.

10) • p. 11, Last paragraph before Section 7: “on a 3-regular graph as the honeycomb lattice” -> “on a 3-regular graph such as the honeycomb lattice”.

Response: Fixed as suggested by the referee.

---

## Editorial Decision

published